# Feasibility and Impact of Adapted Physical Activity (APA) in Cancer Outpatients Beginning Medical Anti-Tumoral Treatment: The UMA-CHAPA Study

**DOI:** 10.3390/cancers14081993

**Published:** 2022-04-14

**Authors:** Amélie Lemoine, Marine Perrier, Camille Mazza, Anne Quinquenel, Mathilde Brasseur, Alain Delmer, Hervé Vallerand, Maxime Dewolf, Eric Bertin, Coralie Barbe, Damien Botsen, Olivier Bouché

**Affiliations:** 1Department of Medical Oncology, Godinot Cancer Institute, 51100 Reims, France; camille.mazza@reims.unicancer.fr (C.M.); damien.botsen@reims.unicancer.fr (D.B.); 2Department of Gastroenterology and Digestive Oncology, CHU Reims, University of Reims Champagne-Ardenne (URCA), 51100 Reims, France; mperrier@chu-reims.fr (M.P.); mbrasseur@chu-reims.fr (M.B.); obouche@chu-reims.fr (O.B.); 3Department of Clinical Hematology, CHU Reims, 51100 Reims, France; aquinquenel@chu-reims.fr (A.Q.); adelmer@chu-reims.fr (A.D.); 4Department of Pulmonary Medicine, CHU Reims, 51100 Reims, France; hvallerand@chu-reims.fr (H.V.); mdewolf@chu-reims.fr (M.D.); 5Department of Nutrition, Endocrinology and Diabetology, CHU Reims, 51100 Reims, France; ebertin@chu-reims.fr; 6Research on Health University Department, University of Reims Champagne-Ardenne (URCA), 51100 Reims, France; coralie.barbe1@univ-reims.fr

**Keywords:** adapted physical activity, feasibility, muscle strength, anxiety, cancer patients

## Abstract

**Simple Summary:**

Physical activity may reduce the risk of overall cancer incidence and improve survival in cancer patients. The beneficial effects of physical activity are also described in cancer survivors but remains poorly known during systemic cancer treatment. Therefore, we studied the feasibility of an adapted physical activity (APA) program in cancer outpatients beginning a medical anti-tumoral treatment for a digestive, lung, hematological, or dermatological cancer. We also studied the impact of APA on fatigue, anxiety, depression, and handgrip strength.

**Abstract:**

Adapted physical activity (APA) improves quality of life and cancer outcomes. The aim of this study was to assess the feasibility of an APA program in outpatients beginning medical anticancer treatment. The secondary objective was to assess the impact of APA on fatigue, anxiety, depression, and handgrip strength (HGS). This prospective study was conducted between January and July 2017. Among 226 patients beginning treatment in the unit for a digestive, lung, hematological, or dermatological cancer, 163 were included. Adherence to the APA program was defined as more than or equal to one one-hour session per week for 3 months. The first evaluation was conducted at 3 months (M3), and the second evaluation at 6 months (M6). A total of 163 patients were included (mean age 62.5 ± 14.3); 139 (85.3%) agreed to follow the APA program. At M3, 106 of them were evaluated, of which 86 (81.1%) declared that they had followed the program. Improvement in anxiety was observed at M3 (−1.0 ± 3.2; *p* = 0.002) but there was no significant change in fatigue or depression. HGS decreased significantly (−1.2 ± 5.5; *p* = 0.04). The APA program was feasible in cancer outpatients beginning medical anticancer treatment. APA should be part of standard support care.

## 1. Introduction

Sarcopenia is characterized by low muscle strength (dynapenia), with low muscle quantity or quality, and/or low physical performance [1]. Primary sarcopenia is defined as a result of the aging process in the absence of any underlying comorbidity, whereas secondary sarcopenia occurs during a chronic inflammatory state (associated with health conditions including cancer) [2]. Some studies have shown a higher risk of chemotherapy toxicity in cancer patients with sarcopenia undergoing chemotherapy [3,4,5]. The negative influence of sarcopenia has been also described on postoperative complications [6] and survival in cancer patients [7,8]. It is now well recognized that sarcopenia is a poor prognostic factor in cancer treatment and outcomes [9].

Patients undergoing active anticancer treatment experience various adverse events, fatigue being the most common [10]. Cancer-related fatigue (CRF) is defined as a cognitive weariness related to cancer and interfering with usual functioning [10,11]. Other symptoms such as anxiety, depression, sleeping disorders, and pain have been described with a potential negative impact on quality of life (QoL) [10].

A growing amount of literature suggests that physical activity (PA) may reduce the risk of overall cancer incidence and improve survival for several cancers [12,13]. Strong evidence for an association between PA and a reduced risk of approximately 10–20% of cancer incidence has recently been described [12,14]. Greater amounts of PA could reduce all-cause and cancer-specific mortality by 40% to 50% among patients with breast, colorectal, or prostate cancer [12,15]. Similarly, overall sedentary behavior has been associated with an increased risk of colon or endometrial cancers [16]. Hence, the World Health Organization (WHO) recommends both regular aerobic and muscle-strengthening activities, as well as reducing sedentary behaviors for optimal health outcomes in the general population [17]. Moreover, the efficacy of rehabilitative interventions has been well described in cancer survivors, by improving functional well-being, QoL, fatigue, anxiety, and depression [18,19,20], and by decreasing risk of cancer recurrence. Notably, these results were confirmed in recent randomized controlled trials including colorectal cancer survivors [21,22].

Only a few previous studies on the benefit of PA have been conducted during surgical and medical anti-tumoral treatments, with most of the data coming from cancer survivors. A preoperative habilitation could be predictive of the postoperative course, by reducing postoperative complications, morbidity, and mortality [23,24,25,26,27]. Practicing a PA during cancer treatment has also been associated with decreased treatment-related toxicities—such as CRF or chemotherapy-induced peripheral neuropathy [28,29,30,31]—and may potentiate the therapeutic effects of certain treatments [32]. Women with breast cancer doing physical exercise during adjuvant chemotherapy and/or radiotherapy experienced a significant reduction in fatigue [33]. Similar results were observed among patients with advanced cancer in the palliative care phase, in which the practice of PA increased their QoL, fitness, and strength, and decreased CRF [34]. All these findings support a regular PA throughout the cancer care continuum as a part of supportive care [35], but the feasibility of PA programs during cancer treatments remains poorly known.

The aim of the UMA-CHAPA study was to assess the feasibility of an adapted physical activity (APA) program in cancer outpatients beginning medical antitumoral treatment for a digestive, lung, hematological, or dermatological cancer. The secondary outcome was to evaluate the impact of this APA program on fatigue, anxiety, depression, and handgrip strength (HGS).

## 2. Materials and Methods

### 2.1. Study Design and Patients

This prospective, descriptive, and monocentric study was conducted in the Chemotherapy Ambulatory Care Unit (UMA-CH) of the Reims University Hospital (France). The study population included adult patients (18 years and older) beginning chemotherapy or targeted therapy at UMA-CH for a digestive, lung, hematological, or dermatological cancer, regardless of cancer staging, between January 2017 and July 2017. For patients who agreed to participate but refused the APA program, only limited data were recorded at inclusion (patient characteristics, pathological features, and causes of refusal) and no follow-up was performed. For patients who agreed to participate in the study and adhere to the APA program, the following data were recorded at inclusion: patient characteristics, comorbidities, nutritional status, pathological features, and the presence and type of previous and actual PA. The first evaluation was conducted between 12 and 16 weeks (M3), and the second evaluation between 24 and 28 weeks (M6). The following data were recorded during the follow-up: ECOG performance status (PS), nutritional status, pathological features, and APA program. In addition, fatigue, anxiety, depression, and muscle strength were assessed at inclusion and during follow-up at M3 and M6.

Nutritional status was assessed by weight, plasma albumin level, and Body Mass Index (BMI, kg/m^2^). Malnutrition was defined as BMI < 18.5 kg/m^2^ in patients aged less than 70 years old, and <21 kg/m^2^ in patients aged more than 70 years old. Exclusion criteria included prior chemotherapy or targeted therapy for the same cancer, or a history of neurological defect that prevented PA. Patients less than 18 years old and/or under guardianship were also excluded.

### 2.2. Ethics Approval

All patients agreeing to participate in the UMA-CHAPA study provided informed written consent. This study was approved by the Ethics Committee (CPP Nord Ouest I, Rouen, 27 January 2017) and retrospectively registered in clinicaltrials.gov (NCT03049436; 10 February 2017).

### 2.3. Exposure Assessment

#### 2.3.1. Multidimensional Fatigue Inventory-20

Fatigue was assessed by the Multidimensional Fatigue Inventory (MFI)-20 scale, which is a reliable and valid international tool to assess fatigue in cancer patients [36,37]. This self-report questionnaire consists of 20 items grouped into five dimensions: general fatigue, physical fatigue, mental fatigue, reduced motivation, and reduced activity. Each item was scored from 1 (“yes, it’s true”) to 5 (“no, it’s wrong”). Each subscale contained four items and scored from 4 to 20. A higher score meant higher fatigue.

#### 2.3.2. Hospital Anxiety and Depression Scale (HADS)

HADS is a questionnaire developed to measure anxiety and depression. This self-report questionnaire includes 14 items, 7 to assess anxiety and 7 for depression. Each subscale (anxiety and depression) was scored from 0 to 21. For each setting, a score of <8 corresponds to an absence of anxiety or depression; a score between 8 to 10 for mild anxiety or depression; a score between 11 to 14 for moderate anxiety or depression; and a score > 15 for severe anxiety or depression [38,39].

#### 2.3.3. Hand Grip Strength (HGS) Measurement

The HGS test was used to measure upper limb muscle strength as an indication of an individual’s overall strength. This test has previously been validated in digestive cancer outpatients [40] and is a reliable and effective tool to screen for sarcopenia and malnutrition [41]. HGS was measured with a hydraulic Jamar dynamometer for each upper limb, both dominant and non-dominant. Patients performed the test while sitting comfortably with both feet touching the ground, hand holding the dynamometer, elbow flexed at 90°, and shoulder adducted. The other upper limb was relaxed and placed alongside the body. Patients were instructed to perform a maximal isometric contraction of three seconds with each hand in turn. Two measurements were taken for each arm. The highest value was retained for the final evaluation. According to the UMA-CHAPA protocol using the EWGSOP definition of sarcopenia in 2010 [42], low muscle strength (dynapenia) was defined as HGS < 30 kg in men and <20 kg in women.

### 2.4. Intervention

#### APA Program

A first meeting with the APA educator was held at patient inclusion. The APA program consisted of at least one one-hour session per week over 3 months, performed at the hospital with a sports coach or outside the hospital with or without a sports coach. Several types of PA could be chosen: aerobic activities (such as Nordic walking, aerobics, running, or swimming), strength training, and relaxation techniques (such as yoga or stretching).

Exercise intensity was measured in metabolic equivalent tasks (METs). One MET was equivalent to the level of basal energy expenditure when sitting on a chair (3.5 mL/O_2_/kg). Activities greater than 2 METs were considered PA. Aerobic activities were ranked between 5 and 10 METs, strength training more than 7 METs, and relaxation less than 3 METs. Other manual or domestic PA such as gardening, DIY, fishing, or housework were ranked less than 3 METs.

### 2.5. Statistical Analysis

Data were described using mean and standard deviation for quantitative variables, and number and percentages for qualitative variables. Univariate analysis (Wilcoxon test, Chi square test, or Fisher exact test, as appropriate) was used to compare patients who agreed to follow the APA program with those who refused. Paired *t*-tests or McNemar tests (as appropriate) were used to assess changes in muscle strength, the presence of dynapenia, MFI-20 score, HADS score, and the presence of depression and/or anxiety between inclusion and during follow-up at M3 and M6. A *p* value < 0.05 was considered significant. Statistical analyses were performed using SAS statistical software version 9.4 (SAS Inc., Cary, NC, USA).

## 3. Results

### 3.1. Patient Characteristics

Between January 2017 and July 2017, 226 cancer outpatients were hospitalized for the first time in UMA-CH. The study was not proposed to 47 patients because of logistical or organizational barriers (lack of time or availability of study recruiters). Among the 179 patients assessed for eligibility, 163 (91.1%) agreed to participate in the study (Figure 1).

The characteristics of the 163 patients included are presented in Table 1. The mean age was 62.5 ± 14.3 years. One hundred forty-two patients (87.1%) had an ECOG PS less than or equal to 1. A total of 65 patients (39.9%) had digestive cancer, 56 (34.4%) had hematological cancer, 26 (16%) had lung cancer, and 16 (9.8%) had dermatological cancer. A total of 143 patients received chemotherapy (87.7%) and 54 (33.1%) received targeted therapy.

Among these 163 patients, 139 agreed to participate in the APA program (85.3%). These patients were significantly younger (61.4 ± 14 years versus 68.7 ± 14.2 years; *p* = 0.03) and more frequently with ECOG PS 0 (n = 49, 35.3% versus n = 1, 4.2%; *p* = 0.002) than the patients who refused the APA program. There was no difference of tumor staging and location between patients who followed the APA program and patients who refused. Among the 24 patients (14.7%) who refused the program, 18 (75%) did not feel capable of doing PA, including 12 patients for reasons of fatigue and 11 patients for pain. Three lacked availability (12.5%), and five indicated no interest (20.8%).

Data collected from the 139 cancer patients who agreed to the APA program are shown in Table 2. Fifty-three of them (38.1%) had comorbidities, mainly high blood pressure (n = 47, 33.8%). The mean BMI was 25.7 ± 5.2 kg/m^2^. Colorectal cancer was the most frequent digestive location (n = 23, 42.6%), and non-Hodgkin’s lymphoma was the most frequent hematological location (n = 21, 42%). All skin cancers were melanomas (n = 15, 100%). Most lung cancers were adenocarcinomas (n = 11, 57.9%). Fifty-three patients (60.2%) were treated for metastatic disease. A total of 88 patients (63.3%) underwent chemotherapy alone, 18 patients (13%) underwent targeted therapy alone, and 33 patients (23.7%) underwent both. At the time of inclusion, 32 patients (23%) had dynapenia. The mean muscle strength was 32.0 ± 11.7 kg in the whole cohort. According to MFI-20, the general fatigue score was 13.1 ± 4.2, the physical fatigue score was 10.7 ± 2.9, the mental fatigue score was 16.2 ± 3.9, the reduced motivation score was 14.5 ± 4.0, and the reduced activity score was 12.2 ± 4.1. The mean anxiety score was 7.1 ± 4.0, and the mean depression score was 4.7 ± 3.6. Fifty-four patients (33.1%) presented anxiety and thirty-two patients (19.6%) presented depression.

Before cancer diagnosis, 130 patients (93.5%) practiced PA, most often with an intensity of >5 METs (n = 101, 77.7%). After diagnosis, 87 patients (62.6%) continued regular PA. Most patients chose to carry out the APA program outside the hospital without a sports coach (n = 95, 69.1%) (Table 3).

### 3.2. Evaluation at 3 Months (M3)

#### 3.2.1. Patient Characteristics

Among the 139 patients who agreed to the APA program, 106 (76.3%) were evaluated at M3. Thirty-three patients (23.7%) were not evaluated at M3: 12 (48%) discontinued treatment, 8 (32%) died, 3 (12.0%) refused evaluation, and 2 (8.0%) were hospitalized and could not be assessed. Thirty-one patients (29.3%) were hospitalized during the 3-month follow-up, 90.3% because of cancer-related causes (toxicity, progression, or side effects).

At M3, 90 patients were still ECOG PS ≤ 1 (86.4%). The mean BMI was 24.9 ± 4.6 kg/m^2^, and nine patients (8.6%) were malnourished. Ninety-four patients (88.7%) were still undergoing chemotherapy and/or targeted therapy. Among them, 16 (17%) had to change treatment because of cancer progression (n = 7, 43.8%), toxicity (n = 1, 6.3%), or maintenance treatment (n = 8, 50%).

#### 3.2.2. Feasibility and Impact

Thirteen patients (12.3%) performed the APA program at the hospital with a sports coach versus 93 (87.7%) outside the hospital, with or without a sports coach. Finally, 86 of the 106 patients (81.1%) declared that they followed the APA program. The main reasons for non-adherence to APA were fatigue and pain. None of the 106 patients stopped the APA program because of a lack of interest.

The evolution of HGS value, MFI-20 scores, and HADS scores at M3 are presented in Table 4. At M3, muscle strength was moderately but significantly lower (mean evolution at M3: −1.2 ± 5.5; *p* = 0.04), but dynapenia was not more frequent (n = 21 (22.3%) at M3 versus n = 32 (23.0%) at inclusion; *p* = 0.90). The HADS anxiety score was significantly lower (mean evolution at M3: −1.0 ± 3.2; *p* = 0.002) than at inclusion, and the presence of anxiety was significantly less frequent (n = 31 (19.0%) at M3 versus n = 54 (33.1%) at inclusion; *p* = 0.004). The intensity of the PA did not influence the evolution of muscle strength nor MFI-20 and HADS scores.

### 3.3. Evaluation at 6 Months (M6)

#### 3.3.1. Patient Characteristics

Among the 139 patients at baseline, 100 patients (71.9%) were evaluated at M6. A total of 39 patients (28.1%) were not assessed at M6: 21 (53.8%) discontinued treatment, 10 (25.6%) died, 3 (7.7%) refused evaluation, and 5 (12.8%) were hospitalized at the evaluation time. A total of 25 patients were hospitalized between M3 and M6, 23 of them (92%) because of cancer-related health issues. The median follow up was 200.5 days (160–282).

At M6, 74 patients were still ECOG PS ≤ 1 (86%). The mean BMI was 24.7 ± 5.3 kg/m^2^. Among these 100 patients, 60 (60%) were still undergoing anti-tumoral treatment.

#### 3.3.2. Feasibility and Impact

The changes in HGS values, MFI-20 scores, and HADS scores at M6 are presented in Table 4. The HADS anxiety score was significantly lower (mean evolution at M6: −1.6 ± 3.8; *p* = 0.0002) than at inclusion, and the presence of anxiety was significantly less frequent (n = 23 (14.1%) at M6 versus n = 54 (33.1%) at inclusion; *p* < 0.0001). The intensity of PA did not influence the change in either muscle strength or MFI-20 and HADS scores.

## 4. Discussion

The present study suggested that APA was feasible in outpatients beginning medical anti-tumoral treatment for digestive, hematological, lung, or dermatological cancer. Among cancer patients who agreed to participate in the APA program, adhesion to the program was 81.1% at 3 months. The main reasons for non-adherence to the APA program were fatigue and pain. Following the APA program significantly reduced anxiety at 3 months and 6 months, but no improvement of HGS was observed. The intensity of PA did not influence the evolution of muscle strength nor MFI-20 and HADS scores.

In the pilot study of Piringer et al. [43], non-adherence was the main obstacle to assess the feasibility of PA after adjuvant treatment in colorectal cancer patients. In our study, the global adherence to the APA program was high, similar to previous studies [34,44,45]. Few previous prospective studies have showed that exercising during anticancer treatment was feasible [28,29,33,34,46], with most data coming from cohorts of cancer survivors. APA in day hospitals has been reported to be possible and relevant among patients with digestive cancer [47], but, interestingly, patients were unaware of the usefulness of APA, suggesting that better informed medical information could improve patient adherence to such a program. Fatigue and pain were the most common reasons for failure to adhere to the APA program. Previous investigations on physical exercise during adjuvant chemotherapy have suggested that non-adherence to an APA program was mainly due to cancer-related symptoms, treatment-related adverse events, or hospital admission [44,48]. The UMA-CHAPA study showed a significant decrease in anxiety, especially at M3, which was consistent with other earlier studies evaluating supervised APA programs with a greater improvement in functional ability and anxiety level in intervention groups [49,50]. However, no significant association between PA and a change in depression or fatigue was established in this study. A recent meta-analysis showed a benefit for QoL, fatigue, aerobic fitness, and muscle strength of PA in the palliative care phase for patients with advanced cancer [34]. Randomized trials are ongoing to evaluate the benefit of APA on health-related QoL in patients undergoing adjuvant (APACaPOp) [51] or palliative chemotherapy (APACaP) [52] for pancreatic cancer.

The pathophysiological mechanisms explaining the potential anticancer benefits of PA are not yet fully elucidated, but certain biochemical pathways have been evoked [14,53,54]. PA could have direct effects on insulin-like growth factor, oxidative stress and antioxidant pathways, epigenetic effects on gene expression and DNA repair, immunity, chronic inflammation and prostaglandins, energy metabolism, or even insulin resistance, leading to a slowing or decrease of tumor growth [53]. Thus, the combination of both regular aerobic training and resistance exercise are recommended [17], and future studies could try to elucidate the recommended doses and intensities of physical exercise among cancer patients. A previous randomized controlled trial conducted by Van Waart et al. suggested that a supervised, moderate- to high-intensity, combined resistance and aerobic exercise program was the most effective for patients undergoing adjuvant chemotherapy for breast cancer [55]. A progressive resistance training in patients with pancreatic cancer significantly improved muscle strength for some muscle groups [56]. Conversely, we found no significant benefit of higher intensities of PA in the UMA-CHAPA study; muscle strength was significantly lower at M3 despite PA. This could be explained by a lack of patient compliance (few sessions performed) or poor adherence to the APA intensity chosen at baseline. The protocol for the proposed APA program included at least one one-hour session per week, for 3 months. The number of sessions performed over 3 months was self-reported for patients who practiced APA outside the hospital. However, supervised physical exercise has been reported to be more effective than home-based training [56]. Conversely, the randomized PASAPAS trial evaluating a 6-month APA program in patients undergoing adjuvant chemotherapy for breast cancer has observed a decrease in compliance with a supervised, moderate- to high-intensity, combined resistance and aerobic exercise program (48% versus 55% for a home-based, low-intensity, individualized, self-managed physical activity program) [55]. It is possible that the home-based APA sessions chosen by most of our patients improved the adherence rate to the UMA-CHAPA program but decreased the intensity of exercises performed. We did not show any differences in feasibility depending on whether APA was performed (in or out of hospital). Future prospective randomized studies could improve compliance by scheduling regular phone calls by caregivers to encourage and assess the adherence, or by using logbooks to self-report the number of home-based APA sessions.

The present study had several limitations. First, the UMA-CHAPA study was not a randomized trial, but the study was conducted prospectively. Another limitation was the missing HGS data at M3 and M6 due to patients lost to follow-up, having completed their anti-tumor treatment before the end of the study. The intensity level (MET) of APA at M3 and M6 was not considered in statistical analyses, but the type of exercise chosen for the program at inclusion was respected. However, the level of APA intensity could have decreased, especially in patients experiencing treatment-related side effects or tumor progression. Moreover, the MFI-20 and HADS scores were subjective and non-exhaustive assessments, and specific cutoffs for cancer patients have not been defined. Other factors may have influenced the observed improvement in anxiety: anxiety could be high at inclusion because patients were starting medical care (unknown location and medical staff, first chemotherapy or targeted therapy session), and the decrease in anxiety at M3 may also be related to tumor control for some patients. Finally, the patient sample was very heterogeneous. Tumor location and treatment intensity were not considered. Perhaps lung cancer patients had more limiting symptoms, such as dyspnea, which could reduce the feasibility of the APA program. It would be interesting to measure these symptoms related to cancer location, to better adapt the type and intensity of physical training according to cancer type, thereby improving the feasibility and benefits of APA. The strength of the study was the prospective evaluation of an APA program in patients with different types of cancer at the onset of anti-tumoral treatment.

Growing evidence suggests that nutritional management and PA are key points for cancer care [13,21,35,46]. Adherence to diet and PA can reduce the risk of overall cancer incidence in the general population. In cancer patients, APA is promoted as soon as the diagnosis is made, in order to improve functional well-being, QoL, fatigue, anxiety, and depression; reduce treatment-related side effects; potentiate the effects of antitumor treatments; and reduce the risk of cancer recurrence [32]. APA programs will constitute truly supportive care complementary of anticancer treatments throughout the cancer care [35]. Further prospective randomized trials are needed to determine the optimal type and intensity of exercises, and to better understand the biochemical mechanisms underlying PA on carcinogenesis.

## 5. Conclusions

APA was routinely feasible in outpatients beginning medical anti-tumoral treatment for digestive, hematological, lung, or dermatological cancer. Following the APA program significantly reduced anxiety at 3 months and 6 months. APA should be part of standard support care. Access and contents have to be further developed and adapted to meet individual needs.

## Figures and Tables

**Figure 1 cancers-14-01993-f001:**
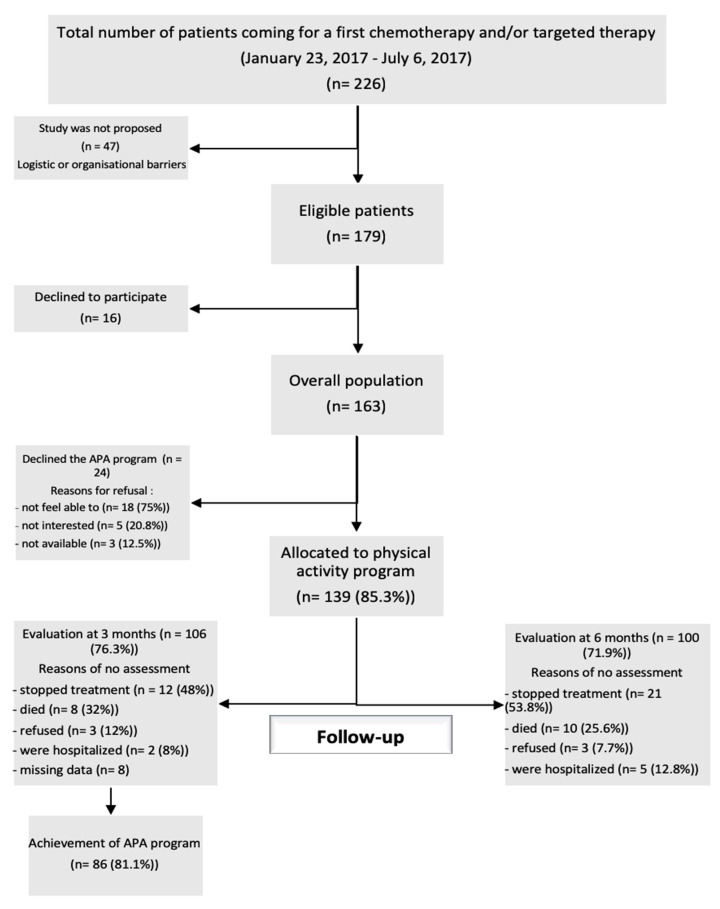
Flowchart of the UMA-CHAPA study.

**Table 1 cancers-14-01993-t001:** Characteristics at baseline of overall population, and patients who agreed or refused to follow the APA program.

Characteristics, n (%)	Overall Population n = 163	Patients Who Agreed to Follow the APA Program n = 139	Patients Who Refused to Follow the APA Program n = 24	*p* Value
**Age (mean ± SD), years**	62.5 ± 14.3	61.4 ± 14.0	68.7 ± 14.2	0.03
**Sex** Male Female	98 (60.1) 65 (39.9)	81 (58.3) 58 (41.7)	17 (70.8) 7 (29.2)	0.25
**ECOG PS** 0 1 2 3	50 (30.7) 92 (56.4) 19 (11.7) 2 (1.2)	49 (35.3) 75 (54.0) 15 (10.8) 0 (0)	1 (4.2) 17 (70.8) 4 (16.7) 2 (8.3)	0.0003 0.002
**Charlson comorbidity index (mean ± SD)**	4.4 ± 2.2	4.3 ± 2.3	4.9 ± 1.8	0.21
**Study proposer** Sport coach Resident Doctor	154 (94.5) 8 (4.9) 1 (0.6)	130 (93.5) 8 (5.8) 1 (0.7)	24 (100) 0 (0) 0 (0)	0.66
**Tumor location** Digestive Hematological Lung **Dermatological**	65 (39.9) 56 (34.4) 26 (16.0) 16 (9.8)	54 (38.9) 50 (36.0) 20 (14.4) 15 (10.8)	11 (45.8) 6 (25.0) 6 (25.0) 1 (4.2)	0.39
**Tumor stage ^1^** Localized Locally advanced Metastatic	23 (21.7) 19 (17.9) 64 (60.4)	20 (22.7) 15 (17.1) 53 (60.2)	3 (16.7) 4 (22.2) 11 (61.1)	0.82
**Chemotherapy** Yes No	143 (87.7) 20 (12.3)	121 (87.0) 18 (13.0)	22 (91.7) 2 (8.3)	0.74
**Targeting therapy** Yes No	54 (33.1) 109 (66.9)	51 (36.7) 88 (63.3)	3 (12.5) 21 (87.5)	0.02

Abbreviations: APA, adapted physical activity; ECOG PS, Eastern Cooperative Oncologic Group Performance Status; SD, standard deviation. ^1^ Fifty-seven missing data corresponding to hematological diseases (lymphoma, leukemia, multiple myeloma) that cannot be classified as solid tumors.

**Table 2 cancers-14-01993-t002:** Characteristics of patients who agreed to perform the APA (n = 139).

Characteristics of Patients, n (%)	Patients Who Agreed to Perform the APA Program n = 139
**Comorbidities** Yes No	53 (38.1) 86 (61.9)
**Type of comorbidity** Stroke Neuropathy Arterial hypertension Coronaropathy Obstructive pneumopathy disease	1 (0.7) 0 (0) 47 (33.8) 4 (2.9) 9 (6.5)
**Nutritional status** **Body Mass Index (mean ± SD, kg/m²)** **Albumin (mean ± SD, g/L) ^1^** **CRP (mean ± SD, g/L) ^2^** **Malnutrition ^3^**	25.7 ± 5.2 38.8 ± 5.3 26.2 ± 47.9 9 (6.5)
**Tumor location** Digestive Hematological Lung **Dermatological**	54 (38.9) 50 (36.0) 20 (14.4) 15 (10.8)
**Tumor stage ^4^** Localized Locally advanced Metastatic	20 (22.7) 15 (17.1) 53 (60.2)
**Previous cancer surgery** Yes **No**	42 (30.2) 97 (69.8)
**Treatment received** Chemotherapy alone Targeting therapy alone Chemotherapy and biotherapy	88 (63.3) 18 (13.0) 33 (23.7)
**Chemotherapy protocol** **Mono-chemotherapy ^5^** **Bi-chemotherapy ^6^** **Tri-chemotherapy or more ^7^**	34 (28.1) 50 (41.3) 37 (30.6)
**Targeted therapy protocol** Bevacizumab Rituximab **Immunotherapy ^8^** Bortezomib **Others ^9^**	7 (13.7) 18 (35.3) 16 (31.4) 5 (9.8) 5 (9.8)
**Indication of treatment ^10^** Adjuvant Neo adjuvant Palliative	17 (17.9) 9 (9.5) 69 (72.6)
**Muscle strength (mean ± SD), kg**	32 ± 11.7
**Dynapenia ^11^** Yes No	32 (23.0) 107 (77.0)
**MFI-20 score (mean ± SD)** General fatigue Physical fatigue Mental fatigue Reduced activity Reduced motivation	13.1 ± 4.1 10.7 ± 2.9 16.2 ± 3.9 12.2 ± 4.1 14.5 ± 4
**HADS score (mean ± SD)** Anxiety Depression	7.1 ± 4 4.7 ± 3.6

Abbreviations: APA, adapted physical activity; CRP, C-reactive protein; ECOG PS, Eastern Cooperative Oncologic Group Performance Status; HADS, Hospital Anxiety and Depression Scale; MFI-20, Multidimensional Fatigue Inventory-20; SD, standard deviation.^1^ Thirty-one missing data; ^2^ twenty-nine missing data; ^3^ malnutrition was defined as BMI < 18.5 kg/m^2^ in patients aged less than 70 years old, and BMI < 21 kg/m^2^ in patients aged more than 70 years old; ^4^ fifty-one missing data corresponding to hematological diseases (lymphoma, leukemia, multiple myeloma) that cannot be classified as solid tumors; ^5^ mono-chemotherapy involved GEMZAR (n = 8), LV5FU2 (n = 7), 5-AZACYTIDINE (n = 5), CYCLOPHOSPHAMIDE (n = 4), MELPHALAN (n = 3), BENDAMUSTINE (n = 3), THALIDOMIDE (n = 1), VINCRISTINE (n = 1), CHLORAMBUCIL (n = 1), and PACLITAXEL (n = 1); ^6^ bi-chemotherapy involved 5FU + OXALIPLATINE (n = 17), CARBOPLATINE + PEMETREXED (n = 9), CARBOPLATINE + PACLITAXEL (n = 6), CARBOPLATINE + ETOPOSIDE (n = 5), 5FU + IRINOTECAN (n = 4), GEMCITABINE + OXALIPLATINE (n = 2), MITOMYCINE + LV5FU2 (n = 2), CISPLATINE + ETOPOSIDE (n = 1), CISPLATINE + GEMZAR (n = 1), 5FU + DETICENE (n = 1), XELODA + OXALIPLATINE (n = 1), and ARACYTINE + DAUNORUBICINE (n = 1); ^7^ tri-chemotherapy or more involved CHOP (n = 13), 5FU + IRINOTECAN + OXALIPLATINE (n = 9), ABVD (n = 7), BEACOPP (n = 4), VTD (n = 2), ACBVP (n = 1), and VINORELBINE + GEMCITABINE + CAELYX (n = 1); ^8^ immunotherapy involved PEMBROLIZUMAB (n = 11), NIVOLUMAB (n = 2), and PEMBROLIZUMAB + NIVOLUMAB (n = 3); ^9^ other targeted therapy involved CETUXIMAB (n = 1), BORTEZOMIB + DARATUMUMAB (n = 1), BRENTUXIMAB (n = 1), OBINUTUZUMAB (n = 1), and OBINUTUZUMAB + IBRUTINIB (n = 1); ^10^ forty-four missing data corresponding to hematological cancers that cannot be staged; ^11^ dynapenia was defined as HGS < 30 kg in men and <20 kg in women according to the UMA-CHAPA protocol using the EWGSOP definition of sarcopenia in 2010.

**Table 3 cancers-14-01993-t003:** Description of physical activity of patients who agreed to perform the APA program.

Physical Activity (PA)	Patients Who Agreed to Perform the APA Program n = 139
**PA before cancer diagnosis** Yes No	130 (93.5) 9 (6.5)
**Intensity of PA before cancer diagnosis** >7 METs >5 METs <3 METs	21 (16.2) 101 (77.7) 8 (6.2)
**PA since cancer diagnosis** Yes No	87 (62.6) 52 (37.4)
**Intensity of PA since cancer diagnosis ^1^** >7 METs >5 METs <3 METs	13 (14.9) 69 (79.3) 5 (5.8)
**Chosen place for APA program** Hospital Outside hospital with a sports coach Outside hospital without a sports coach	19 (13.7) 25 (18.0) 95 (68.4)
**Chosen intensity for APA program** >7 METs >5 METs <3 METs	40 (28.8) 96 (69.1) 3 (2.2)

Abbreviations: APA, adapted physical activity; MET, metabolic equivalent task; PA, physical activity. ^1^ Fifty-two missing data.

**Table 4 cancers-14-01993-t004:** Change in muscle strength, MFI-20, and HADS scores at M3 and M6.

Characteristics of Patients (Mean ± SD)	At Baseline (n = 139)	M3 (n = 106)	M6 (n = 100)
Muscle strength, kg	32.0 ± 11.7	31.9 ± 11.7 *	31.2 ± 12.2
MFI-20 score			
General fatigue	13.1 ± 4.1	12.5 ± 3.9	12.2 ± 4.5
Physical fatigue	10.7 ± 2.9	11.1 ± 2.8	11.1 ± 3.0
Mental fatigue	16.2 ± 3.9	16.3 ± 3.5	15.3 ± 4.1
Reduced activity	12.2 ± 4.1	13.0 ± 3.6	12.5 ± 4.3
Reduced motivation	14.5 ± 4	14.8 ± 3.4	14.4 ± 4.1
HADS score			
Anxiety	7.1 ± 4.0	5.9 ± 3.7 *	5.7 ± 3.2 *
Depression	4.7 ± 3.6	4.6 ± 3.4	5.1 ± 4.2

* *p* < 0.05 (comparison with baseline). Abbreviations: HADS, Hospital Anxiety and Depression Scale; MFI-20, Multidimensional Fatigue Inventory-20; M3, month 3; M6, month 6; SD, standard deviation.

## Data Availability

The data presented in this study are available upon request from the corresponding author.

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
