# Peer review of "Feasibility and Impact of Adapted Physical Activity (APA) in Cancer Outpatients Beginning Medical Anti-Tumoral Treatment: The UMA-CHAPA Study"

_cancers, 2022, doi:10.3390/cancers14081993_

Round 1
Reviewer 1 Report
Maintenance or increase of physical activity is an almost daily subject in the management of cancer patients. The group of patients ist very heterogenous in this regard. It ranges from physically very active individuals to individuals with high BMI, who should have started 20 years ago. In addition, the underlying malignant diseases and treatment intensity are very different in the study population. Thus it's not surprising, that changes in fatigue and depression were not significantly reduced.
Although scientifically sound, this heterogeneity limits the significance of the study. Thus my conclusion would be more likley: ... APA should be part of standard support care. Access and contents have to be further developed and adapted to meet individual needs.
Author Response
Dear Reviewer,
Thank you for your review report and your constructive comments.
As you notified, another limitation of the study was the heterogeneity of the patient sample. Indeed, patient tumor location and stage, treatment intensity, nutritional status and comorbidities may have influenced the impact on fatigue, anxiety or depression.
Therefore we edited a part of discussion and conclusion as you suggested:
Lines 429-430 : “Finally, patient sample was very heterogeneous. Tumor location and treatment intensity were not considered”.
Lines 448-452 : “APA was routinely feasible in outpatients beginning medical anti-tumoral treatment for digestive, hematological, lung or dermatological cancer. Following the APA program significantly reduced anxiety at 3 months and 6 months, but no improvement of de-pression or HGS strength was observed. APA should be part of standard support care. Access and contents have to be further developed and adapted to meet individual needs”.
Kind regards,
Dr A. Lemoine
Reviewer 2 Report
I think the study raises a very important issue that is often disregarded in daily clinical practice. I read the paper with interest.
The strength is the big sample size and I believe conducting such investigation is not that easy due to the non-adherence in general.
Methods and results are very clearly described. Limitations were pointed correctly.
The majority of references come from the last 5 years, which is a great advantage, although they are entirely not prepared according to Journal's instructions - it has to be improved.
Author Response
Dear Reviewer,
Thank you very much for your report.
As you recommended, we improved the form of the references according to Journal's instructions.
Kind regards,
Dr A. Lemoine
Reviewer 3 Report
Feasibility and Impact of Adapted Physical Activity (APA) in Cancer Outpatients Beginning Medical Anti-tumoral Treatment: The UMA-CHAPA study
Amélie Lemoine et al aimed to assess the feasibility of an Adapted Physical Activity (APA) program in cancer outpatients beginning medical antitumoral treatment for a digestive, lung, hematological or dermatological cancer.
This is an interesting and well-written paper about the feasibility of an APA program for cancer patients.
I have just a few comments that could help to improve the manuscript as regards readability
- Please move the paragraph “Aims” to the last line of the Introduction section
- Please structure the paper as follows: Introduction, Material, and Methods: Study design and patients, Ethics, Exposure Assessment (Multidimensional Fatigue Inventory, HADS, HGS), Intervention (APA Program), Statistical Analysis (please state the type of t-test used)
- In Table 2, there is repeated information (the first four variables are described in Table 1
- I think authors should discuss a bit more about the differences between patients who agreed and not to participate in the study and the use of a t-test
And finally, a question, why did the authors choose the t-test if there are many statistical methods that can better reflect the temporal dimension of the study.
Author Response
Dear Reviewer,
Thank you for the interest you show in our study, your comments and constructive remarks.
Point 1: Please move the paragraph “Aims” to the last line of the Introduction section. Please structure the paper as follows: Introduction, Material, and Methods: Study design and patients, Ethics, Exposure Assessment (Multidimensional Fatigue Inventory, HADS, HGS), Intervention (APA Program), Statistical Analysis
Response1 : We moved the paragraph “Aims” to the last line of the Introduction section and structure “Materials and Methods” as you suggested: Study design and patients, Ethics approval, Exposure assessment, Intervention and Statistical analysis.
Lines 86-90: “The aim of the UMA-CHAPA study was to assess the feasibility of an Adapted Physical Activity (APA) program in cancer outpatients beginning medical antitumoral treatment for a digestive, lung, hematological or dermatological cancer. The secondary outcome was to evaluate the impact of this APA program on fatigue, anxiety, depression and handgrip strength (HGS)”.
Point 2: In Table 2, there is repeated information (the first four variables are described in Table 1
Response 2: We deleted the first four variables of Table 2 to avoid repetitions with Table 1 (Line 270).
Point 3: In Statistical Analysis, please state the type of t-test used
Response 3: We specified the t-test used in the Statistical analysis section:
Lines 203-212: “Data were described using mean and standard deviation for quantitative variables, and number and percentages for qualitative variables. Univariate analysis (Wilcoxon test, Chi square test or Fisher exact test, as appropriate) was used to compare patients who agreed to follow the APA program with those who refused. Paired t tests or McNemar tests (as appropriate) were used to assess changes in muscle strength, presence of dynapenia, MFI-20 score, HADS score, presence of depression and/or anxiety between inclusion and during follow-up at M3 and M6. A p value < 0.05 was considered significant. Statistical analyzes were performed using SAS statistical software version 9.4 (SAS Inc., Cary, NC)”.
Point 4: According to the difference between patients who agreed and not to participate in the study and the use of a t-test, and why we choose the t-test if there are many statistical methods that can better reflect the temporal dimension of the study :
Response 4: We did not perform repeated measures ANOVA for two reasons:
- Pragmatically, a loss of patients was expected (death, treatment discontinuation, withdrawal…) at 3 and 6 months. The repeated measures ANOVA would consider only patients with the three measurement times resulting in loss of power.
- The objective of the study was to compare the characteristics of the patients at 3 and 6 months with their initial characteristics. The repeated measures ANOVA would only provide an analysis of the overall time effect without identifying a particular time.
Thank you very much for your consideration.
Dr A. Lemoine